# Technical, Economic, and Intelligent Optimization for the Optimal Sizing of a Hybrid Renewable Energy System with a Multi Storage System on Remote Island in Tunisia

Mohamed Hajjaji [1,2,*] , Dhafer Mezghani [1], Christian Cristofari [2] and Abdelkader Mami [1]

1    UR-LAPER Laboratory, Faculty of Sciences of Tunis, University of Tunis El Manar, Tunis 1068, Tunisia
2    UMR 6134 SPE Laboratory, University of Corsica, 20250 Corte, France
*    Correspondence: hajajji.mohamed@fst.utm.tn

**Abstract:** Due to their small dimension and isolated energy systems, islands face a significant energy supply challenge. In general, they use fossil fuels for electricity generation. Fossil fuels are a major source of $CO_2$ emissions, and they are very costly. The cost of electricity generation on islands is up to 10 times higher than on the mainland. This situation without a doubt represents a financial burden for the islanders. Using renewable sources, especially solar and wind sources, offers great potential for power generation in remote locations, as they are a clean and inexhaustible source of energy. Electrifying these zones with a hybrid system consisting of a photovoltaic (PV) and wind systems associated to a hydraulic and an electrochemical storage system is a promising alternative. The purpose of this study is to optimize the dimension of the components generation of systems, especially for a remote island in Tunisia. The first part of this object outlines the PV-wind-battery-hydraulic generation system architecture and modeling. The optimal sizing of the device additives, satisfying two criteria with the aid of evolutionary algorithms NSGAII, is defined inside the second part of this article. The outcomes are discussed from the point of view of the importance of the system dimension and in terms of compliance with the criteria through the study of the most optimal particular configurations.

**Keywords:** photovoltaic systems; wind; islanders; evolutionary algorithms; hydraulic storage system; energetic modeling; NSGAII

## 1. Introduction and Background of This Research

Currently, the electrification of islands has become an effective instrument for the sustainable development of these regions both in developing countries and in developed countries. A lot of research has been carried out in the field of renewable energy use on remote islands to increase the penetration of renewable energy [1,2]. More and more islands have become self-sufficient in energy thanks to renewable energy. For example, the island of EL Hierro in Spain is 100% self-sufficient due to their hydro-wind electricity generation system. In addition, the Greek island of Tilos is the first in the Mediterranean to produce almost all of its own electricity from wind and solar energy [3]. Thanks to 11 wind turbines and a biomass power plant, the island of Samso in Denmark was also the first to get completely rid of fossil fuels. However, compared to the number of islands in the world, only 1% of the islands have become self-sufficient in energy. The cost and power system flexibility are among the reasons for the observed low penetration of renewable energies in remote areas. With strong social and environmental incentives to integrate maximum renewable content, improved power system efficiency and reduced system cost can be seen as a pathway to increased renewable energy penetration.

In Tunisia, 21 islands are not populated. They are remote islands. Power shortages and high conventional power cost remains one of the biggest drawbacks of covering remote areas. Islands need a continuous and reliable electricity supply from renewable sources,

especially photovoltaic (PV) sources and wind systems (WT). Exploiting these sources can bring great socio-economic benefits to islands. According to the National Institute of Meteorology, Tunisia has one of the largest solar deposits in the world. The annual average of sunshine exceeds 3000 h; it is also the most important of all the Mediterranean Basin with $1800/m^2/year$. The wind resource is also important, with an annual speed wind equal to 6 m/s [4,5]. However, their intermittent and random character does not guarantee a continuous supply. For this reason, attaching a storage system ($H_2$ storage) can overcome this problem, but this solution is still hampered by the high owning costs [6–12]. Thus, the use of hydraulic storage and a battery energy system associated with the PV/wind system is an attractive concept to provide consumers on remote islands with reliable and cheap electricity. However, this alternative depends upon the optimal combination and sizing design of the renewable energy generation system coupled with the storage system. Recently, different sizing and optimization methodologies have been developed by the research community in order to appreciate the feasibility and the economic effect of such solutions [13–19]. Some studies [20–24] have developed several methods to extract the optimal sizing of renewable systems associated with different storage devices. In Refs. [25,26], Particle Swarm Optimization (PSO) has been used for optimal sizing of hybrid renewable energy systems. In Ref. [27], the authors used the cuckoo optimization algorithm for sizing wind, PV, batteries, and diesel generator components for the lowest cost of energy and loss of load probability. Another study solved the sizing problem by using the grey wolf optimization (GWO) [28]. However, the most common is by far the class of genetic algorithms (GA), and the most widely used one is NSGA-II [29,30]. More recent research has used NSGA II for the optimal sizing of a hybrid renewable energy system [31–33].

In the literature, the demand for electricity on remote islands is usually provided by diesel generators, leading to considerable uncertainty because of the cost of fuel and the high pollution of the surrounding environment [34]. The optimal sizing of PV/wind/hydraulic/battery with multi-objective formulation with combination of cost-LPSP is rarely found in literature.

In this context, this paper presents techno-economic analysis and intelligent optimization of a PV/wind system with hydraulic and battery storage system, ensuring the continuous supply of electricity to a load demand on a remote island in Tunisia.

## 2. Description of the System

Figure 1 presents the proposed architecture of the hybrid PV/wind electricity production system and the hydraulic/electrochemical (bank battery) storage system investigated in this study. Wind energy production system(wind turbine), PV energy production system (PV arrays),bank battery, and hydraulic storage (pump, turbines/generator, and reservoir) are interfaced by static converters in a maximum power point tracking (MPPT) operation, thus enabling energy on the DC bus (48 V) routed to the AC load demand by means of the inverter. The energy flows of the storage system are bidirectional, while they are unidirectional for the production system. The phase of pumping seawater to the reservoir and charging the bank battery when the PV/wind hybrid electricity production has a surplus of energy compared to the energy demand and the turbinated phase and discharge in the event of a lack of energy by the hydraulic turbine and the battery ensures continuity of supply to the remote site. Simulations will be carried out with solar radiation, wind, and load of charge in the remote site, over a year on average, sampling every half an hour. This work is interested in optimizing the dimension of this system based on two contradictory criteria. The crucial phase will consist in defining these criteria for optimizing the problem.

The flowchart of the hybrid PV/wind and hydraulic /battery multi-storage system operation is illustrated in Figure 2. Primary, wind profile, solar irradiation profile, and load demand of the remote site are included. The wind will turn into wind power (Pwt), from which the power generated by the wind turbine will be calculated. In addition, the PV power generated by the PV panels will be measured through a solar irradiation. The sum of PV power (Ppv) and wind power (Pwt) will be compared with power consumption (Pload)

through the DC bus (Pbus). Second, the state of charge of the reservoir (SOCres) and of the battery bank (SOCbat) will be calculated. If Pload < (Ppv + Pwt), we have a surplus of energy, leading to verify the SOCres. If SOCres < SOCresmax is confirmed, the pumping phase until the water tank is full. When, the SOCres = SOCres max and SOCbat < SOCbatmax, the surplus of energy will charge the bank of battery until SOCbat = SOCbatmax. The second condition is when Pload > (Ppv + Pwt). Here we have a deficiency of energy. The system switches to turn on the turbine, which operates until SOCres = SOCresmin. When this condition is verified, the battery discharges and supplies the site with energy until SOCbat = SOCbatmin.

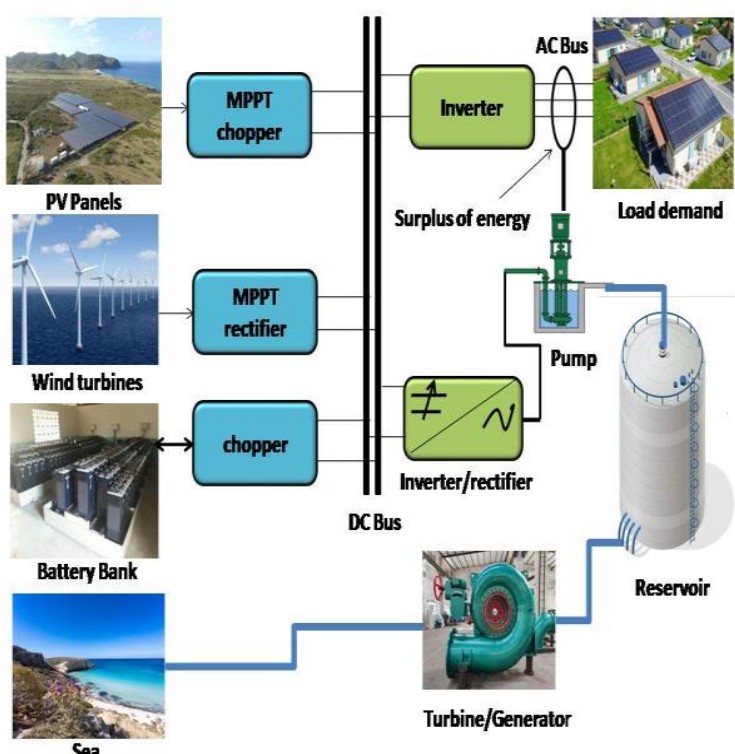

**Figure 1.** System architecture to optimize.

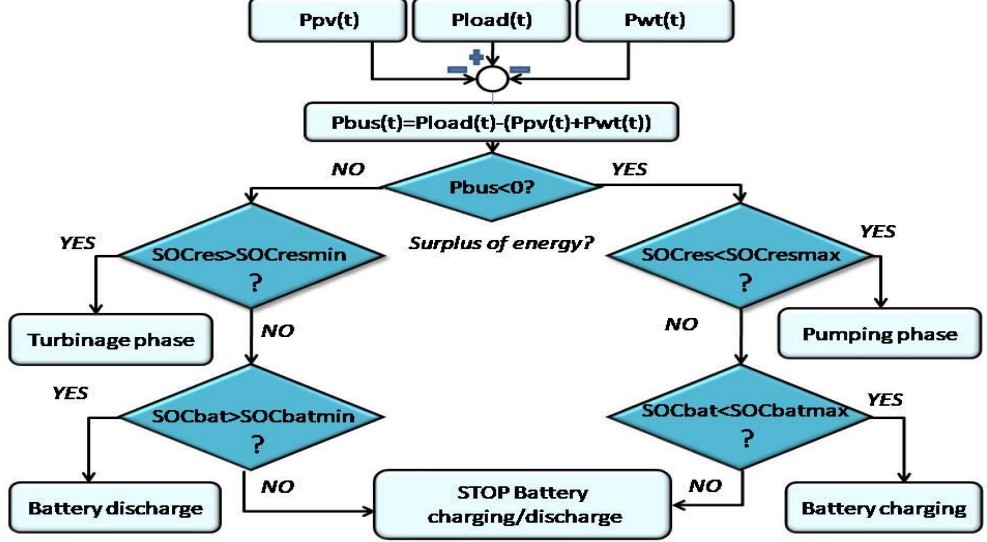

**Figure 2.** Flowchart of the hybrid PV/wind hydraulic/battery multi storage system operation.

## 3. Modeling and Simulation

In this study, we consider that the bus DC voltage is constant and equal to 48 V. All the elements of the system such as PV power source, wind power source, the water storage system, and the bank of battery are coupled to the bus DC through converters (choppers and inverters).These models will be used by the optimal design process. Due to the higher computation time in this type of optimization problems, we use energetic models. The processing time cost is a priority in our work.

### 3.1. Data Capture

The wind speed profile represented in Figure 3 has been obtained from a statistical distribution model (based on Weibull distribution) from the wind energy potential at a typical region in Tunisia. This model successfully models the probability of occurrence of wind resource wind speeds [35,36].

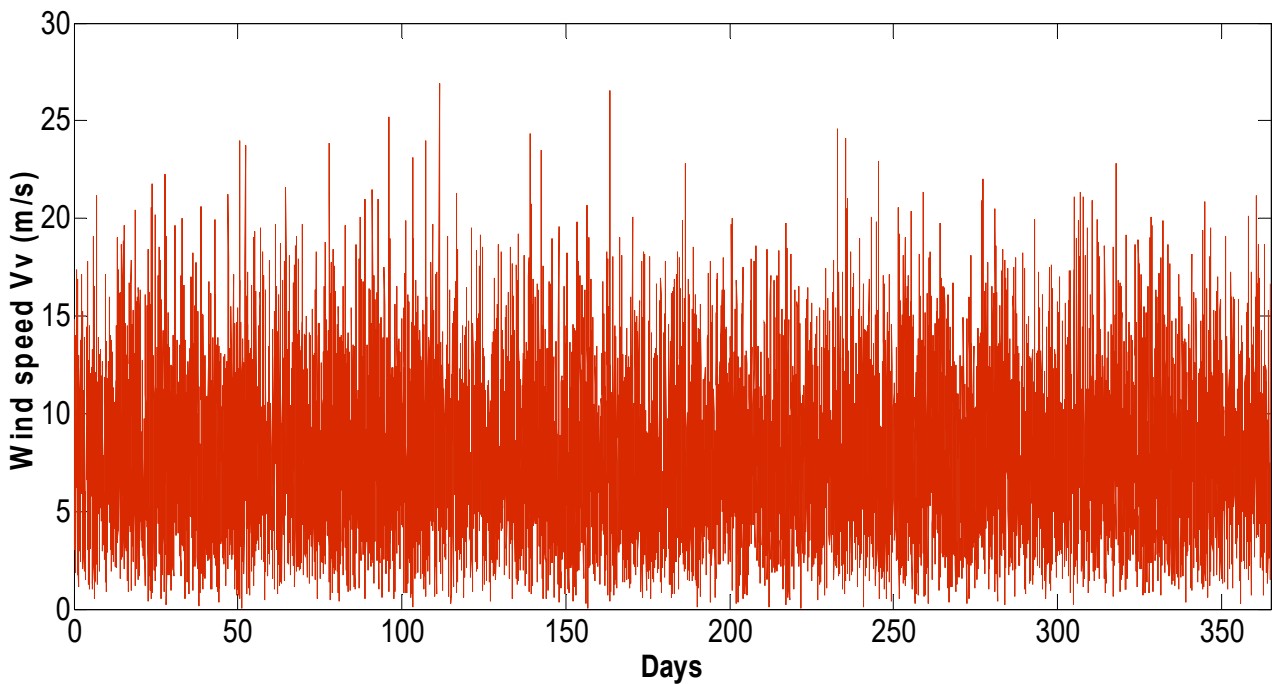

**Figure 3.** Wind speed profile during a year.

The load profile is obtained by estimating the energy needs on a remote site and the behavior of the habitat during the day. The first phase is between 0 h and 5 h 30 min, corresponding to a period during which the demand for electrical energy is low (no activities). The second phase is between 5 h 30 min and 10 h, corresponding to a period of high family activity (lighting, heating, etc.) and agricultural activities (operation of rural equipment): during this phase, electricity consumption increases, with a peak around 7:30 a.m., then it decreases. The third phase, which is approximately between 10 a.m. and 4.45 p.m., corresponds to a period of slower activity due to lunch breaks and interruptions in agricultural activities: during this phase, electricity consumption is at an average level and is quite stable. The fourth phase, which extends from 4:45 p.m. to midnight, corresponds to the return period and the end of agricultural activities; the families return to their homes (lighting, audiovisual, etc.). During this period, the demand for electrical energy is the highest. Electricity consumption increases sharply, and then falls again at the end of the evening, with the peak of consumption being around 6:30 p.m. The description of the lead demand is illustrated by Figure 4.

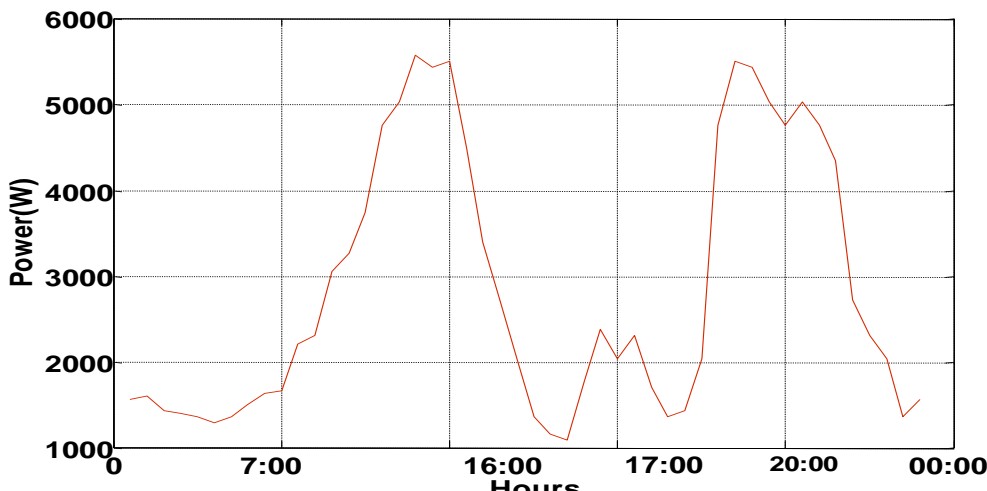

**Figure 4.** Consumption profile for atypical day.

The solar irradiation profile is generated in a deterministic manner from the data, and measurements are done by the National Institute of Meteorology, using high-precision instruments to estimate solar irradiation. Figure 5 represents the power of solar irradiation given every half hour for (sampling time Ts) a typical site in Tunisia over a whole year.

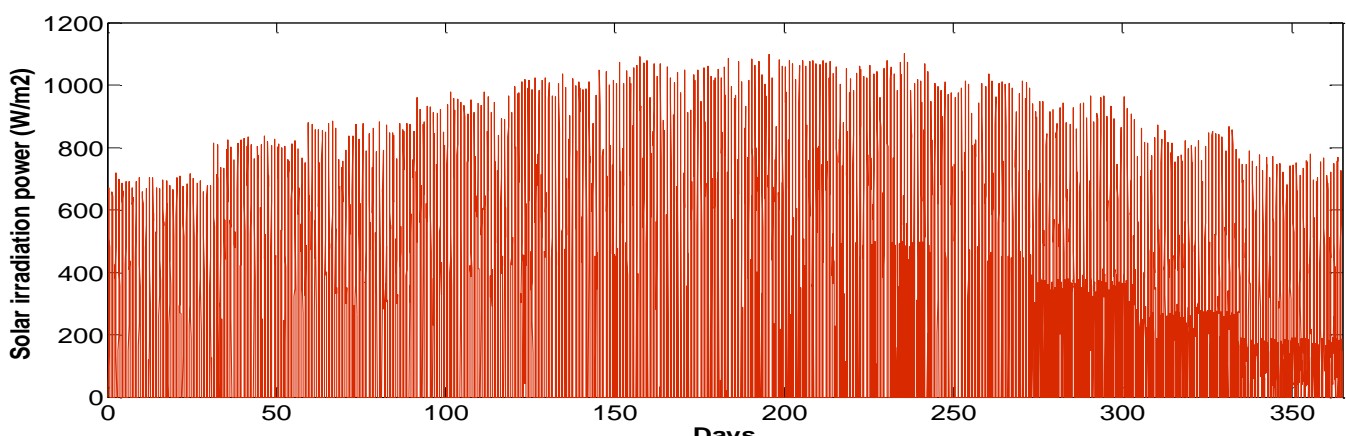

**Figure 5.** Solar irradiation during a year.

*3.2. Wind Turbine Modeling*

In our study, we used an active wind turbine(Air X 400 W) operating at optimal powers through a MPPT rectifier(maximum power points tracker rectifier). This convertor is modeled using an efficiency equal to 95%. The energetic model used in the optimization problem is defined by the electrical power Pwt, which represents the following:

$$P_e = C_p P_v = \frac{1}{2} C_{pmax}.\rho.\pi.R^2.v^3, \tag{1}$$

where $R$ is the blade radius is the air density, and $Cp$ is the power coefficient from manufacturer data corresponding to the considered turbine.

The average value of this power generated by the wind speed over a period of one year, with a sampling interval of half an hour, is equal to 283.3 W, as shown in Figure 6.

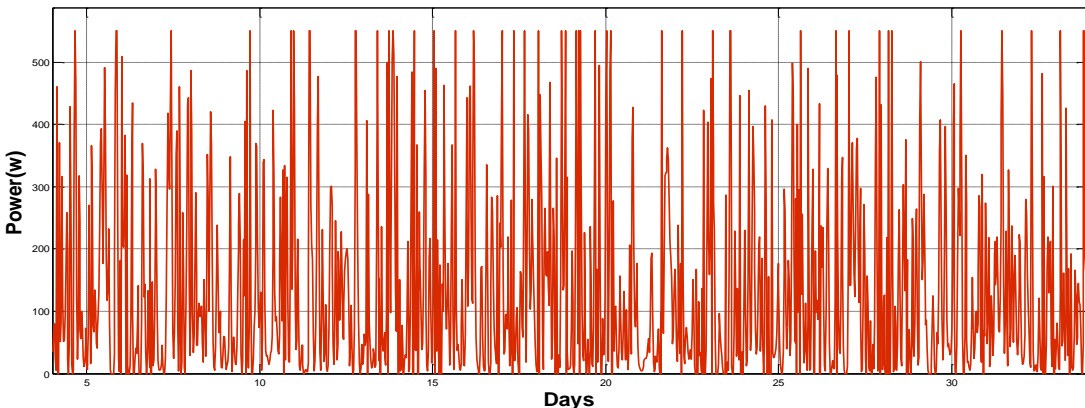

**Figure 6.** Power of wind turbines (w).

*3.3. PV Module Modeling*

In the literature, many models of PV modules have been developed [37]. The famous form is double diode mode, but the existence of a double exponential and six parameters to makes it very complex. In contrast, using a model based on a single diode of the PV generator makes it simple, especially in the case of optimization problems. In our study, we implement the single diode model, or the four-parameter models for simplicity. Then, we will use the model of a PV cell to assimilate it into a module under different lighting and temperature conditions based on the parameters provided by the manufacturers' data sheets.

3.3.1. Environmental Factors That Affect the Performance of PV

Different factors can include pressure, temperature, and many more, but the most critical factors that affect the performance of PV are:

- Solar radiation intensity and spectrum (since the last depends on air mass), effectively received by the module (in W/m$^2$), which is an environmental factor. This effectively received solar radiation can be limited by dust, snow, or any other natural or artificial shadowing;
- Ambient temperature;
- Rainfall;
- Accumulation of dust on the surface of the solar panel [38,39].

3.3.2. Single Diode Model of the PV Generator

It should be noted that this four-parameter model is only valid for mono polycrystalline PV cells. The PV module considered in our study" Sun module SW 250 mono" is deduced from the manufacturer's data. Table 1 summarizes the different characteristics of this PV panel.

The four-parameter model is a model that considers the PV module as an irradiance-dependent current source, connected in parallel with a diode and in series with a series resistor as shown in Figure 7.

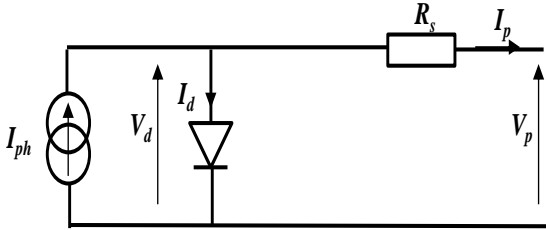

**Figure 7.** Single-diode electrical model (four parameters) of a PV cell.

**Table 1.** Characteristics of the PV module.

| Maximum power:$P_{max}$ | 250 W |
|---|---|
| Maximum power point voltage: $U_{mpp}$ | 31.1 V |
| Open circuit voltage: $V_{oc}$ | 37.8 V |
| Maximum power point current: $I_{mpp}$ | 8.05 A |
| Short circuit current: $I_{sc}$ | 8.28 |
| Temperature coefficient of the short-circuit | 47 °C |
| current: $CT_{Isc}$ | 0.042%/°C |
| Cells per module $N_s$ | 60 |
| Surface: $S_{PV}$ | 1.67 m$^2$ |
| Efficiency: $\eta_{const}$ | 14% |

The current source generated a current $I_{ph}$ proportional to the solar irradiation. The current $I_{ph}$ supplied by the PV cell is expressed by the following equation:

$$I_p = I_{ph} - I_d. \tag{2}$$

The diode current $I_d$ has a given exponential classical form expressed by:

$$I_d = I_S \left[ \exp\left( \frac{V_p + R_s I_p}{n.V_T} \right) - 1 \right]. \tag{3}$$

where $V_T = \frac{k.T_c}{q}$ is the thermodynamic potential (or thermal voltage), $T_c$ is the cell temperature, $I_S$ is the reverse saturation current of the diode, $k$ is the Boltzmann constant ($k = 1.38.10{-}23$ J/K), $q$ is the charge of the electron ($1.6 \times 10^{-19}$ C, and n is the ideality factor of the junction ($n = 1$ for an ideal diode and it is between 1 and 2 depending on the technology). Neglecting the effect of parallel resistance (Rsh), the PV cell current can then be written as:

$$I_p = I_{ph} - I_S \left[ \exp\left( q \frac{V_p + R_s I_p}{n.k.T_c} \right) - 1 \right]. \tag{4}$$

With

$$I_S = I_{Sref}.\left( \frac{T_c}{T_{ref}} \right)^{\frac{3}{n}} \exp\left( \frac{q.E_g}{n.k}.\left( \frac{1}{T_{ref}} - \frac{1}{T_c} \right) \right), \tag{5}$$

where $E_g$ is the band energy (1.12 eV for Si), $T_{ref}$ is the temperature under reference conditions, and $T_c$ is the cell temperature in Kelvin (K).

The photo-current is linked to the illumination $E$, to the temperature $T_c$ and to the photo-current measured under reference conditions by:

$$I_{ph}(E, T) = \left( \frac{E_c}{E_{ref}} \right)\left( I_{phref} + CT_{Isc}.\left( T_c - T_{ref} \right) \right), \tag{6}$$

where $E$ is the illumination in W/m$^2$, $I_{phref}$ is the photo-current under the reference conditions (to simplify the calculation of $I_{ph}$, we often make the approximation that the current $I_{phref}$ is equal to the short-circuit current ISC, ref of the module), and $CT_{Isc}$ is the coefficient of the temperature of the short-circuit current (A/K).

The module temperature $T_c$ varies according to the ambient temperature $T_{amb}$ and the illumination according to the following relationship [34,35].

$$T_c = T_{amb} + \left( \frac{NOCT - 20}{800} \right)E. \tag{7}$$

By dividing this equation by $dI_p$, we will have:

$$R_s = -\frac{dV_p}{dI_p} - \frac{n.k.T}{q.I_S.\exp\left(q\frac{V_p + R_s I_p}{n.k.T}\right)}, \tag{8}$$

$$R_s = -\frac{dV_p}{dI_p}\bigg|_{V_{c0}} - \frac{n.k.T}{q.I_S.\exp\left(q\frac{V_{C0}}{n.k.T}\right)}. \tag{9}$$

Through the fine modeling of the photovoltaic cell, we determine the four parameters of the electrical circuit.

The validation of the fine modeling is illustrated by the following figures.

Figure 8 shows the variations of power and current as a function of voltage for different levels of solar irradiation at a temperature kept constant (T = 25 °C), for the PV panel with maximum power Pppm = 250 W measured under standard test conditions (Est = 1000 W/m², Tst = 25 °C). We note that when the sunshine increases, the intensity of the current increases, with the increasing allowing the module to produce a greater electrical power.

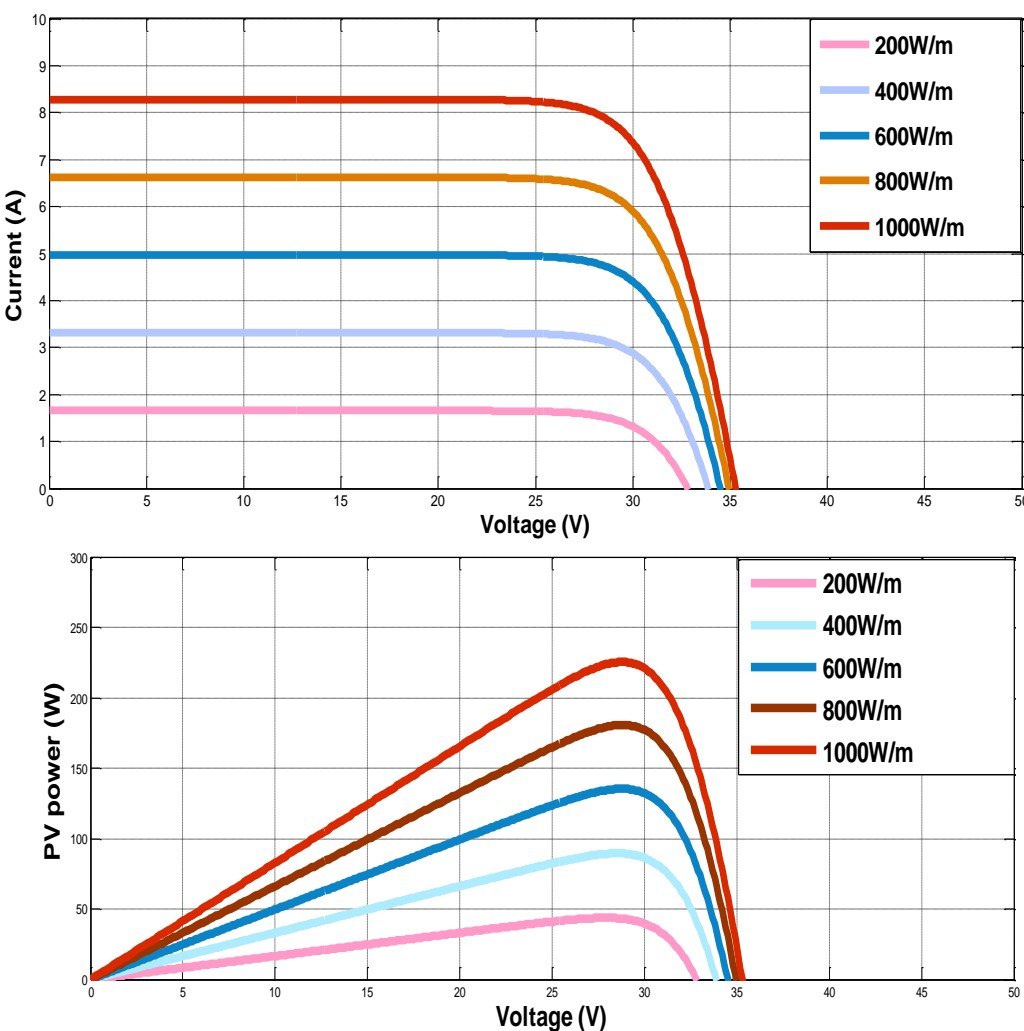

**Figure 8.** Electrical characteristics of the Sun module SW 250 mono panel for different solar irradiations.

Figure 9 shows the effect of temperature change at constant solar irradiation G = 1000 W/m², for the same PV panel. The evolution of the I (V) characteristic as a function of temperature shows that the current increases very rapidly when the temperature

rises, with the same occurring for the P (V) characteristic, as the voltage increases when the temperature rises.

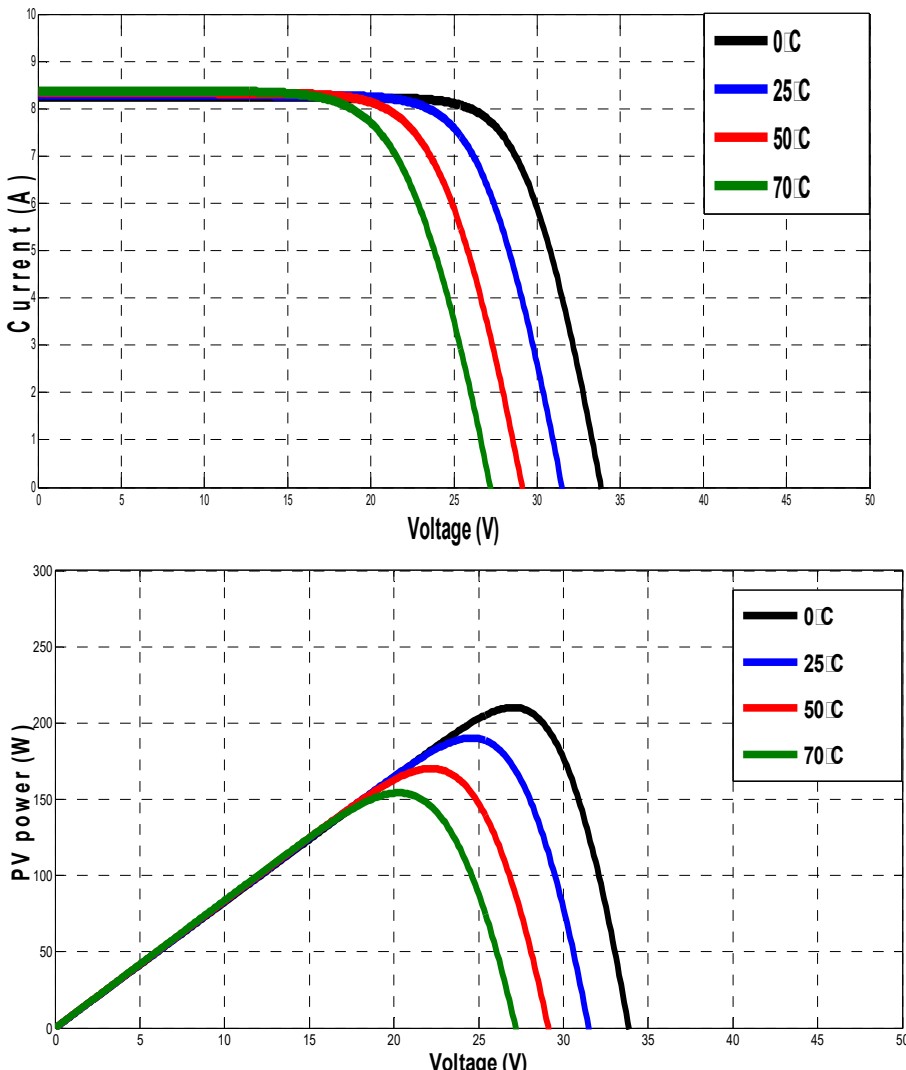

**Figure 9.** Electrical characteristics of the Sun module SW 250 mono panel for different temperatures.

*3.4. Energetic PV Modules*

Previously, we detailed how the fine modeling precisely predicts the behavior of PV panels. However, this modeling is very expensive in terms of time of the digital processor calculation when an optimization process is carried out. In our study, reducing the cost of time is a priority. For that reason, the use of an energy modeling that takes into account that the maximum power regions depending on solar irradiation and temperature is inevitable. In our work, we assume that the PV system is equipped a MPPT device to extract the maximum power.

Therefore, we can model the PV generation of a PV panel by the following simplified model:

$$Pv = S_{PV}.\eta_{PV}.Es. \tag{10}$$

where $S_{PV}$ is the PV panel surface in m$^2$ and $\eta_{PV}$ is the PV panel and PV conversion system (choppers with MPPT) efficiency, which is equal to 10% in this study.

Figure 10 represents the power of PV panels given every half hour for (sampling time Ts) a typical site in Tunisia over a whole year.

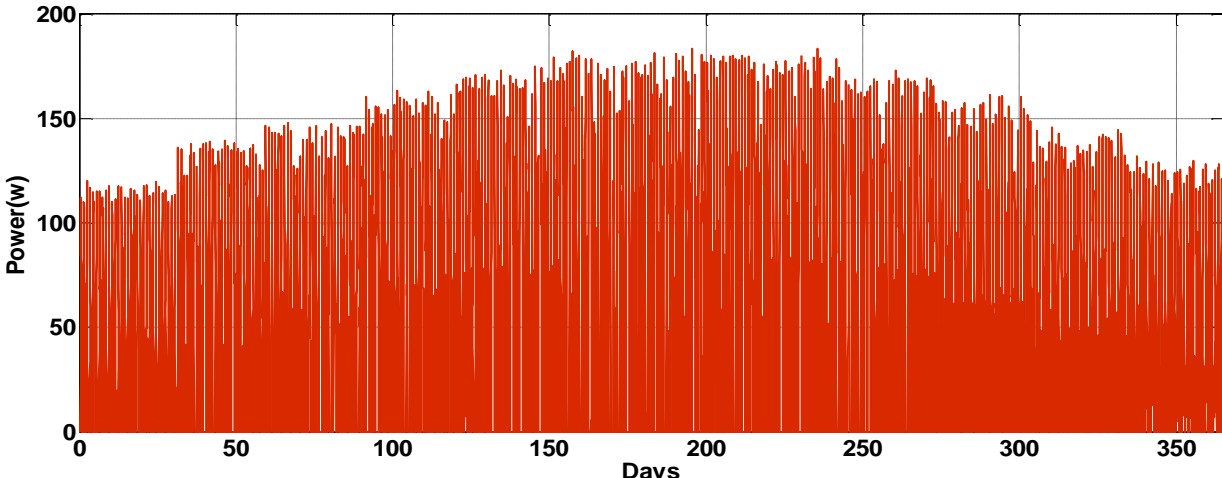

**Figure 10.** Power of the PV panels (w).

*3.5. Hydraulic Storage Model*

In the literature, the two most important variables for sizing the pumping system are the elevated head of the tank and the sea (Hm), and the volume of the water reservoir (V). We consider these variables in the optimization sizing of our system.

In hydraulic storage, we have two phases. The first phase is in the case of excess power of the generator source. This phase is a pumping phase. It can be modulated by:

$$Qpump = PIM/(\rho * g * Hm * \eta p), \tag{11}$$

where the useful power is $P_{IM}$, delivered on the output shaft of the engine as the electromagnetic power efficiency $\eta_{IM}$, which is taken as 95%: and the efficiency of the pump $\eta p$, which is considered equal to 80%.

The second phase is when we have a lack of energy at the load demand. The discharge of the tank operates the turbine to meet the needs of the load. They can be modulated by:

$$Ptur = \eta T * \rho * g * Hm * QT. \tag{12}$$

*3.6. Battery Model*

Our choice was to use lead batteries, given their relatively low cost compared to other technologies. Our modeling is based on a Yuasa NP 38-12I (38A.h 12 V) reference battery. One of the important parameters for the continuation of the study is the value of the capacity for a discharge of 3 h: C3 = 30.3 A.h, as well as the coefficient of Peukert, in which one deduces from the various measurements of the discharge: *n* = 1.28.

The battery is built by assembling elementary cells, in series and/or in parallel. The equivalent diagram of the battery, shown in the figure below, is then deduced from the assembly of the elementary cells using Thévenin's theorem. The current in a cell, used to calculate the state of charge, depends on the type of association made, and is expressed by:

$$I_{cel} = \frac{I_{bat}}{N_{cel\_p}}. \tag{13}$$

The parameters $e_0$ and $r_{cel}$ are the functions of the level of energy available in the battery, of the current delivered, and of the temperature of the accumulator.

In its electrical model Figure 11, the resistance $r_{cel}$ and the voltage $e_0$ both vary depending on the state of charge SOC of the battery. Figure 12 shows the evolution of $r_{cel}$ and $e_0$ depending on the state of charge of a cell.

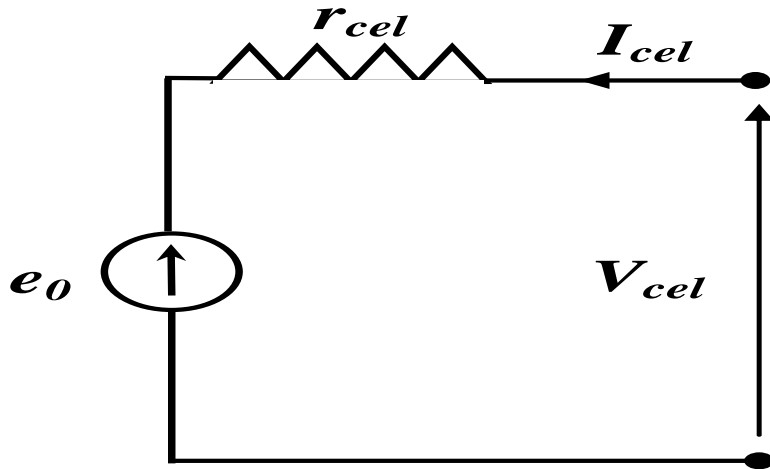

**Figure 11.** Equivalent electrical diagram of an elementary battery cell.

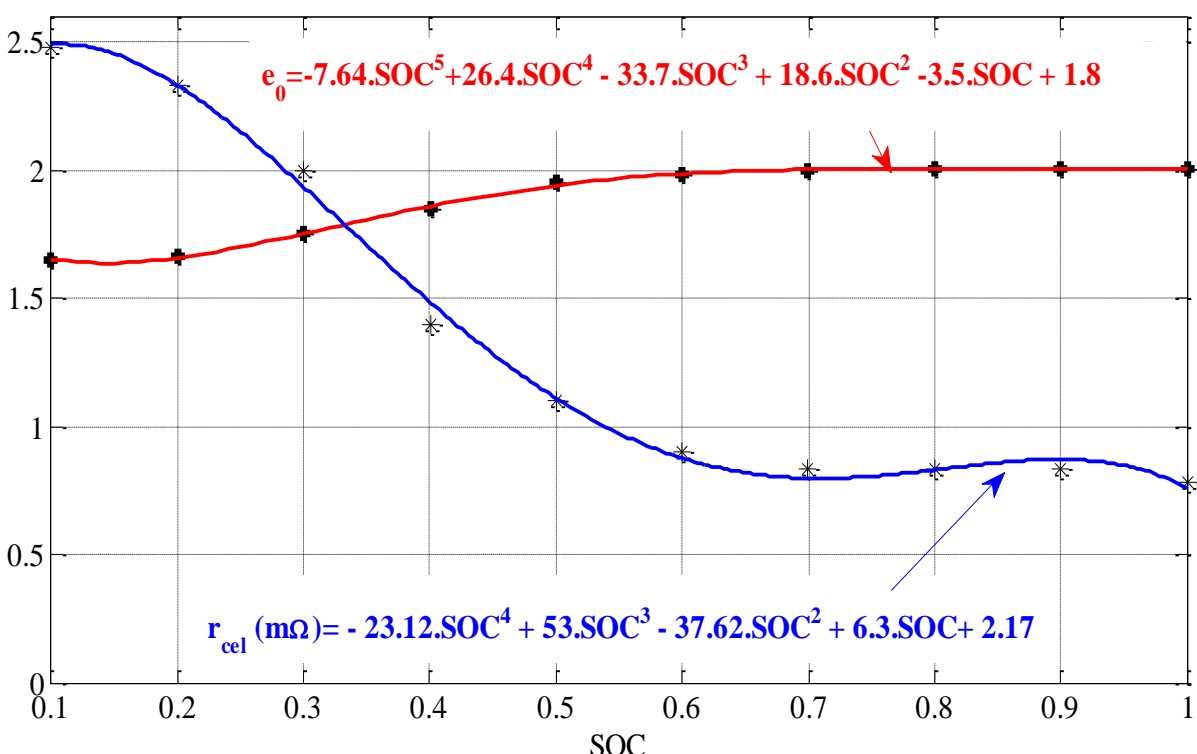

**Figure 12.** Evolution of rcel and $e_0$ as a function of the state of charge of a cell [35].

Factors Influencing the Aging of the Battery

The main factors influencing the aging of a battery are:

- The number of cycles (charge/discharge);
- Temperature, one of the most influencing factors on battery life. Usage at high temperature will accelerate aging due to an increased parasitic reaction. Many studies show that the aging of a battery is divided by 2 to 3 between a usage at 25 °C and the same usage at 45 °C [40].

We suppose in our work that the influence of thermal and aging is not considered in this model for reasons of simplification.

The expression of the voltage DC is:

$$U_{DC} = N_{cel\_s}.e_0(SOC) + \frac{N_{cel\_s}}{N_{cel\_p}} r_{cel}(SOC).I_{bat}.$$ (14)

where $I_{bat} = I_{DC} - I_{conso}$, where $I_{conso}$ designates the consumption current.

Knowledge of the state of charge of a cell *SOC* is a determining element in relation to the behavior of the complete system. One must be able to evaluate it to check if the conversion chain can carry out its mission.

A cell is, from an energy point of view, characterized by its capacity $C_{cel}$. This is the quantity of electricity, indicated in A.h, that it is able to restore after a full charge, and when discharged with a current kept constant. This capacity varies according to several factors such as the intensity of the discharge, the temperature, and the concentration of the electrolyte. Thus, the maximum quantity of electricity available, under a discharge current *I*, is lower than the theoretical capacity of the accumulator, defined for a discharge at infinitesimal current. The quantity of electricity accessible for a discharge in *i* hours at constant current $I_i$ is deduced from the maximum capacity by the empirical relationship of Peukert, which is written as:

$$C_i = C_3 \left( \frac{I_i}{I_3} \right)^{1-n}.$$ (15)

For a discharge at constant current $I_{cel}$, we express the state of charge *SOC* of an elementary cell of the accumulator as follows:

$$SOC(t) = 1 - \frac{I_{cel}}{C_i} \times t.$$ (16)

For our application, the current is constantly variable over time. We then discredited the previous equation by considering the constant current between two calculation steps. We can thus determine the expression of the variation of the state of charge $\Delta SOC_k$ of the cell at time $k.\Delta t$:

$$\Delta SOC_k = \frac{I_{cel_k}}{C_i} \Delta t = \frac{I_{cel_k}}{C_3} \left( \frac{I_{cel_k}}{I_3} \right)^{n-1} \Delta t.$$ (17)

This approach also makes it possible to take into account the battery recharging phases. Indeed, if the current in the cell becomes negative, its state of charge increases. In the end, the state of charge of the cell is expressed by:

$$SOC_k = SOC_{k-1} + \Delta SOC_k.$$ (18)

For optimal operation, management of the state of charge of the battery is considered for limit values of the state of charge of the battery such as $SOC_{batmin} \geq 20\%$ and $SOC_{batmax} \leq 95\%$ [41–43].

## 4. Formulation of the Optimization Problem

Optimization consists in finding the minimum of an optimization criterion (an objective function to find according to a vector $x = (x_1, x_2, \ldots, x_n)^T$, representing the input variables of the optimal solution of *f* [44–47].

### 4.1. Input Variables

The five input variables associated with the system sizing model are chosen as parameters to be optimized. These variables are given by the following Table 2, where their range of variation is also specified.

**Table 2.** Input variables.

| Input Variable: x | *Minimum Value* $x_{min}$ | *Maximum Value* $x_{max}$ |
|---|---|---|
| **Number of batteries: Nbat** | 1 | 100 |
| **Number of wind turbines: Nwt** | 1 | 30 |
| **Number of PV panels: $N_{pv}$** | 1 | 100 |
| **Elevating head: Hm** | 1 | 40 |
| **Water reservoir volume: V** | 1 | 100 |

*4.2. Design Constraints*

To ensure the efficiency of the system, when the design parameters *x* vary over the whole exploration domain, it is necessary to introduce some constraints $g_i$, which can generally be translated into inequalities of the form:

$$g_i(x) \leq 0,$$

where *x* denotes the vector associated with the input variables.

In our study, the chosen constraints are related to the state of charge of the battery (SOCbat) and the tank (SOCres).

For the reservoir, it is crucial to ensure that $SOC_{res}$ does not reach the tolerable limit value, which is equal to $SOC_{resmax}$ = 1 (full tank) and does not fall below to $SOC_{resmin}$ = 0, so that the pump does not work in a vacuum and avoids related problems:

$$SOC_{resmin} - SOC_{res} \leq 0 \text{ and } SOC_{res} - SOC_{resmax} \leq 0.$$

In addition, it necessary to make sure that the battery does not operate in deep discharge and stays in safe state-of-charge zones.

$$SOC_{batmin} - SOC_{bat} \leq 0 \text{ and } SOC_{bat} - SOC_{batmax} \leq 0.$$

*4.3. Optimization Criteria*

The criteria to be optimized can have constraints on their variables through inequalities and vector functions g and h. mathematically, the optimization problem can be formulated as follows:

$$\begin{cases} \min f(x) \\ x_{min} \leq x \leq x_{max}, \quad x \in \mathbb{R}^n \\ g(x) \leq 0 \ et \ h(x) = 0, \ g \in \mathbb{R}^k et \ h \in \mathbb{R}^l \end{cases}. \tag{19}$$

In our study, the optimization problem has two contradictory objectives; the first objective is to minimize the satisfaction of the electrical load demand, known as LPSP (loss of power supply probability). The second objective is to minimize the cost of the $Cost_{sys}$ system over a life cycle of 25 years. Thus, we have a bi-objective optimization problem [48].

*4.4. System Cost Modeling*

In our study, we estimate the cost of energy (cost per Khw) produced by our system through the costs of PV panels, wind turbines, converters, installation, maintenance, and replacement of components over a cycle of life (Tcycle) equal to 25 years. Cost modeling is done through an energy model that relates the cost of each element to its average power.

For the battery cost and for an operating time Tcycle of 25 years and a tank cycle time T of 356 days, the approximate cost of the battery pack over 25 years is expressed by the following relationship:

$$Cot_{bat}[\text{k€}] = N_{cel\_s} \cdot N_{cel\_p} \cdot C_0 \times 10^{-3} \cdot \cdot N_{cy}^{\tau} \cdot \frac{\tau_{exp}}{\tau}. \tag{20}$$

where $C_0$ is the cost of a battery cycle estimated at EUR0.1 (additional batteries). A YUASA 12V battery costs Cbat = EUR108.A 2 V cell costs Célé = 108 EUR/6. We have 180 deep cycles. One deep cycle on a cell costs Célé/180 = EUR 0.1. For the turbine system, the cost is estimated at EUR 50/kW (Cost$_{tur}$). For the pumping system, the cost is estimated at 240EUR/kW(Cost $_{pump}$). The cost, combining the PV panels, converters, cables, sensors, and control circuits, is estimated at EUR 2000/kW(Cost$_{PV}$). For the cost of the wind turbine, the approximate cost model is given by the following equation [8–11]:

$$\text{Cost}_{wt}(\text{KEUR€}) = 1.7 \times P_{wt} + 6$$

where $P_{wt}$ is the average power of a wind turbine.

Finally, the cost model of the Cost$_{sys}$ system can be expressed as follows:

$$\text{Cost}_{sys} = \text{Cost}_{PV} + \text{Cost}_{tur} + \text{Cost}_{bat} + \text{Cost}_{WT} + \text{Cost}_{pump}. \tag{21}$$

### 4.5. Satisfying the Electrical Load LPSP (Loss of Power Supply Probability) Criteria

*LPSP* is the sum of the energy deficits supplied to the load (Loss of Power Supply LPS), encountered during a period $T$ ($T$ = 1 year) and reduced to the annual energy supplied to the annual load ($E_{Load}$) [49].

$$LPSP = \frac{\sum_{t=1}^{T} LPS(T)}{E_{Load_{year}}}. \tag{22}$$

*LPS* represents the difference between the energy demanded by the load $E_{load}$ and the available energy supplied by the generation system and the storage system ($E_{ren}(t) = E_{PV}(t) + E_{WT}(t) + E_{Bat}(t) + ETank\,(t)$), where $E_{PV}$ is the available photovoltaic energy, $E_{WT}$ is the available wind energy, $E_{Bat}$ is the useful energy stored and available in the batteries, and $E_{tank}$ is the useful energy stored in the reservoir, described by the relationship below:

$$LPS(t) = \begin{cases} E_{Load}(t) - E_{ren}(t) & \forall\ E_{Load}(t) < E_{ren}(t) \\ 0 & \forall\ E_{Load}(t) \geq E_{ren}(t) \end{cases}. \tag{23}$$

### 4.6. Optimization Process

Dimension optimization has become a standard approach for the design systems [50–52].

In our study we seek to adapt the five design parameters Npv, Nwt, Nbat, Hm, and V through technical and economic criteria to satisfy the need of the load demand at any time. We used the NSGAII algorithm to adapt these parameters. In several works, NSGAII has shown its efficiency in solving the multi-objective optimization problem. This is why we chose it. The optimization process integrating the NSGA-II algorithm as well as the simulation model of the system is illustrated by the synoptic of the following Figure 13.

The discrete variables NPV, NWT, Nbat, Hm, and V are decoded using the integer function. The typical values of the NSGAII regulation parameters used for the system optimization considered are summarized in Table 3.

**Table 3.** Parameters regulation for NSGAII.

| Settings | Values |
| --- | --- |
| Number of generations | 100 |
| Number of individuals | 100 |
| Number of executions | 3 |
| Variable mutation rate | 0.35 |
| Crossover gene mutation rate | 0.04 |

The evolution of the population as the genetic algorithm is executed is presented in Figure 14, according to the different steps of the NSGA-II algorithm described below.

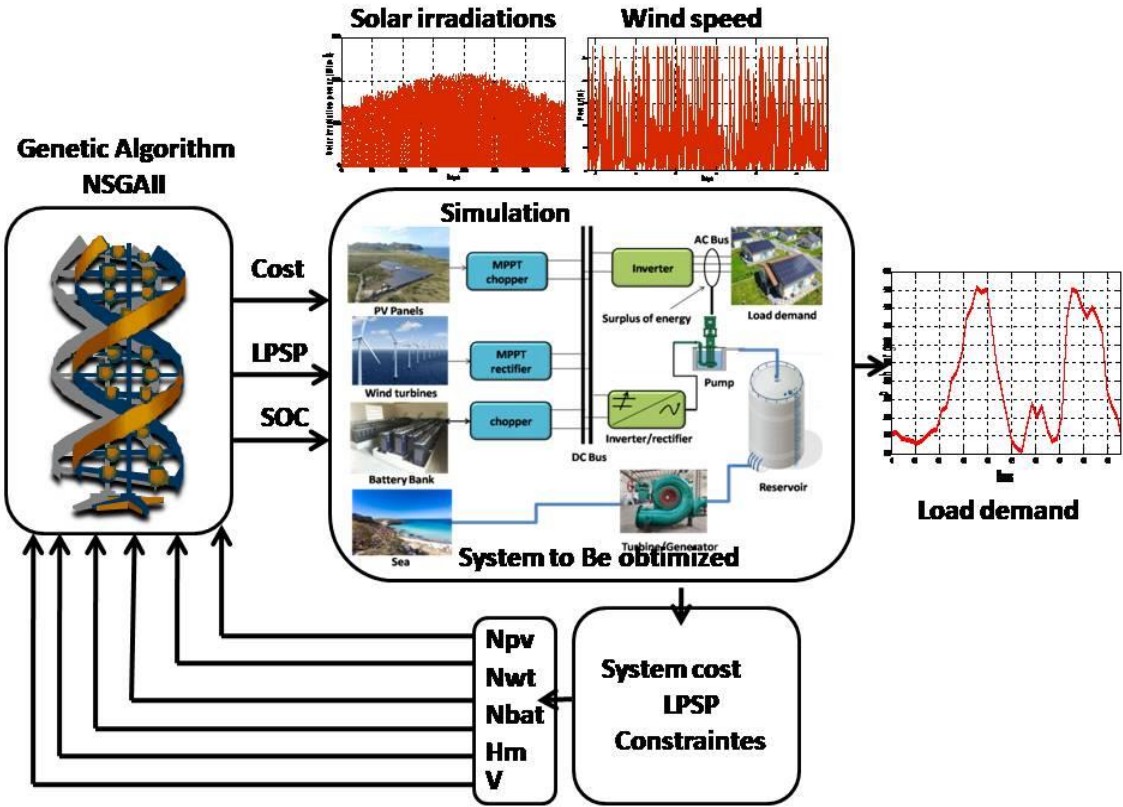

**Figure 13.** Optimization process.

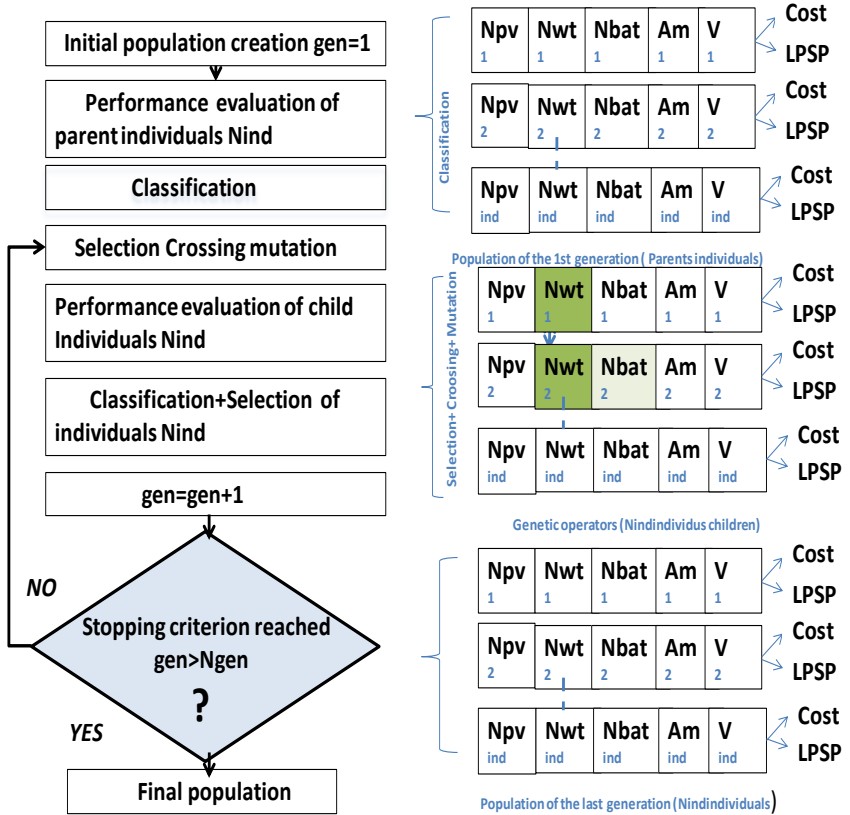

**Figure 14.** Evolution of the population of individuals during the execution of the NSGA-II genetic algorithm.

An initial population of Nind individuals is first randomly created, in order to be used for the 1st generation; each generation evaluates the performances of the 2Nind individuals: the parent Nind from the previous generation, to which Nind children resulting from the application of genetic operators are added. These operators bring a diversification of the individuals at each generation, in order to explore the space of the solutions. Two types of genetic operators (also called evolution operators) are applied to the best performing individuals of the parent population (selection phase).The first genetic operator is crossover. Here the chromosomes of two individuals (two parents) are exchanged so as to create two new individuals (two children), i.e., two new sizes. The second genetic operator is mutation. Here, one of the chromosomes of an individual is modified, resulting in a new dimension. A selection of Nind individuals is then made among the 2Nind individuals according to the Pareto dominance: the dominant individuals are kept (the elitist approach).The population is then completed if necessary with the best remaining individuals in order to obtain a new population of Nind individuals used in the next generation. Thus, the best individuals are never lost from one generation to another [53].

To verify the reproducibility of the results obtained, we carried out three executions of the algorithm with MATLAB R2013a.The numbers of individuals and generations considered result in a computing time of approximately 6 h on an Intel Core i7 processor.

## 5. Results and Discussion

In this part, we present the optimization design process. Our objective is to extract the optimal sizing choice of the system.

Figure 15 presents the pareto front of the two optimization criteria of our problem (cost-LPSP).This front is obtained after three executions. The zone of good compromise circled by blue dots on the optimal front presents the best solutions of our study. By moving outside to this zone, the solutions become insignificant given the dominance of one criterion over another.

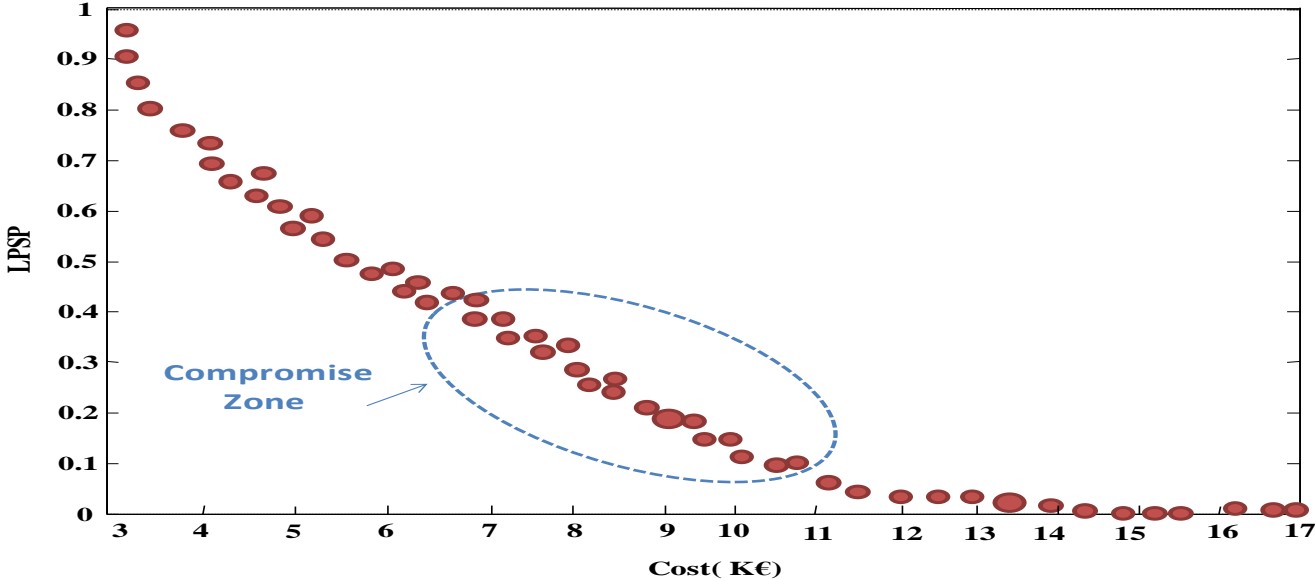

**Figure 15.** Pareto front of optimal system configurations.

The path of the input variables along the optimal front according to the two criteria is illustrated in Figure 16. These results explain how trends of increasing values of input variables are in the direction of inflating the cost of the system. On the other hand, these tendencies are reversed if the increase becomes the direction of LPSP aggravation.

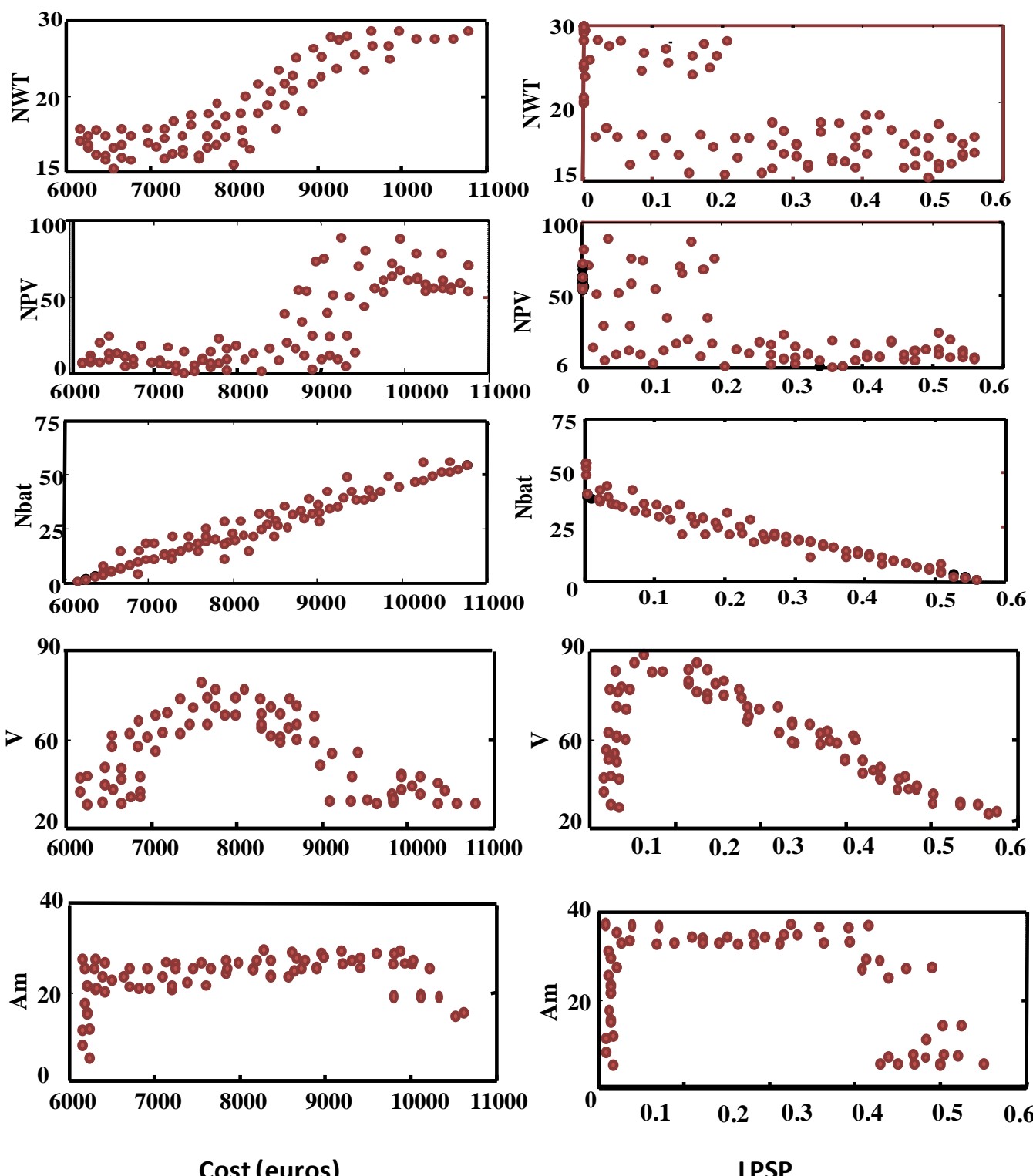

**Figure 16.** Evolution of design parameters along the optimal front as a function of system cost and as a function of LPSP.

To decide on an optimal configuration among this panoply of solutions on the optimal front, it is fundamental to analyze the characteristics of at least three different configurations (Optimum1, Optimum2, and Optimum3) extracted from the optimal front, positioned in the zone of good compromise mentioned above, and whose positions have been articulated in Figure 17.

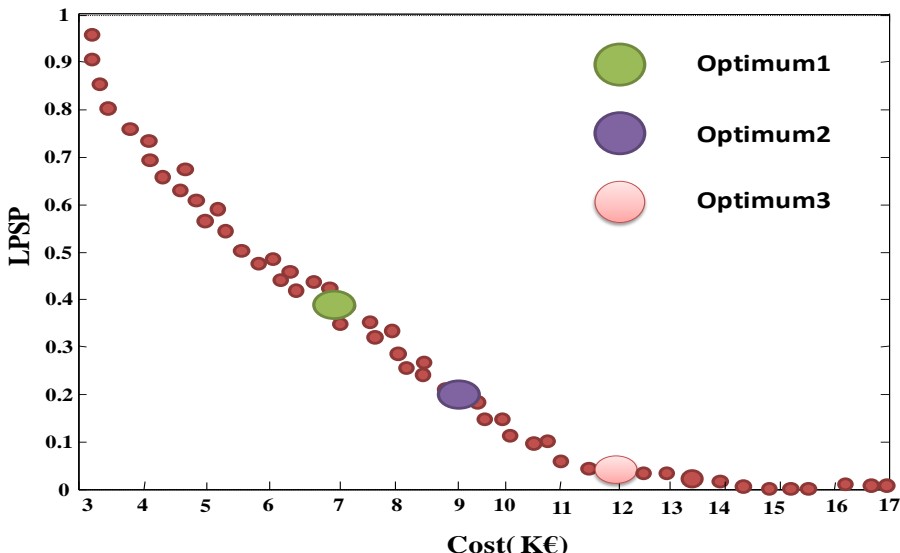

**Figure 17.** Particulate configurations from the good compromise zone of the optimal front.

The characteristics of these configurations are summarized in Table 4.

**Table 4.** The characteristics of the selected configurations.

| Configurations | Optimum1 | Optimum2 | Optimum3 |
|---|---|---|---|
| $N_{bat}$ | 10 | 20 | 50 |
| $N_{WT}$ | 15 | 20 | 28 |
| $N_{PV}$ | 60 | 25 | 20 |
| Am | 22 | 25 | 30 |
| V | 60 | 50 | 40 |
| Cost system ($f_1$) kEUR | 7.1 | 9.2 | 11.1 |
| LPSP ($f_2$) | 0.4 | 0.2 | 0.5 |

The cost of the system presents remarkable differences. Between the first configuration and the second there is a gain about EUR 2000 in terms of system cost, while the LPSP is 40% for the first configuration encountered, and 20% for the second. When we compare the second configuration and the third, there is a notable difference in the cost of the system of about EUR 4000, but the LPSP is 15%. The state of the charge of the battery is represented in Figure 18.If we compare the first configuration and the second one, we note that for the second configuration the battery does not undergo a deep discharge throughout the year SOCbat > 80%, while for the first configuration it decreases up to 30%.As a final point, these interpretations will then be a good marker to choose the correct and the performed configuration of our site.

The results of the simulation for the preferred configuration (Optimum 2) are presented below. We choose two typical days: one in summer and the other in winter.

The objective of this illustration is to demonstrate the complementarity between the different elements of our system to satisfy the load demand at all times.

Figure 19 represents respectively the powers evolution for two typical days: one in summer and the other in winter. In summer, the power delivered by the photovoltaic panels Ppv is important; it reaches 1100 w, which exceeds the need for our site. The surplus of energy therefore goes in the direction of charging the batteries and pumping water into the reservoirs. In winter, Ppv does not exceed 6000 w, which causes the discharging of the turbines and battery t and ensuring the continuity of supply to our site.

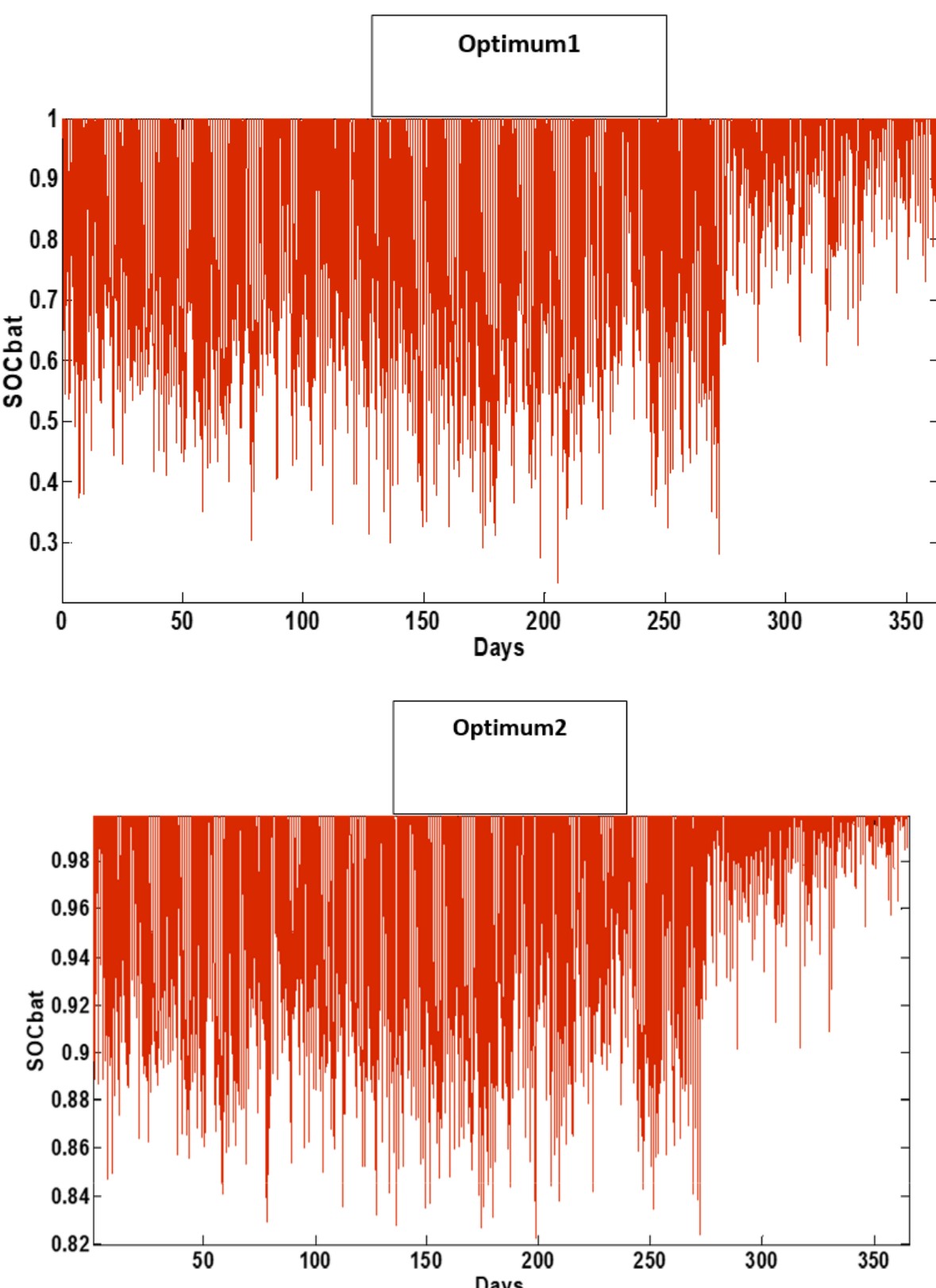

**Figure 18.** Battery SOCbat variations for configurations 1 and 2.

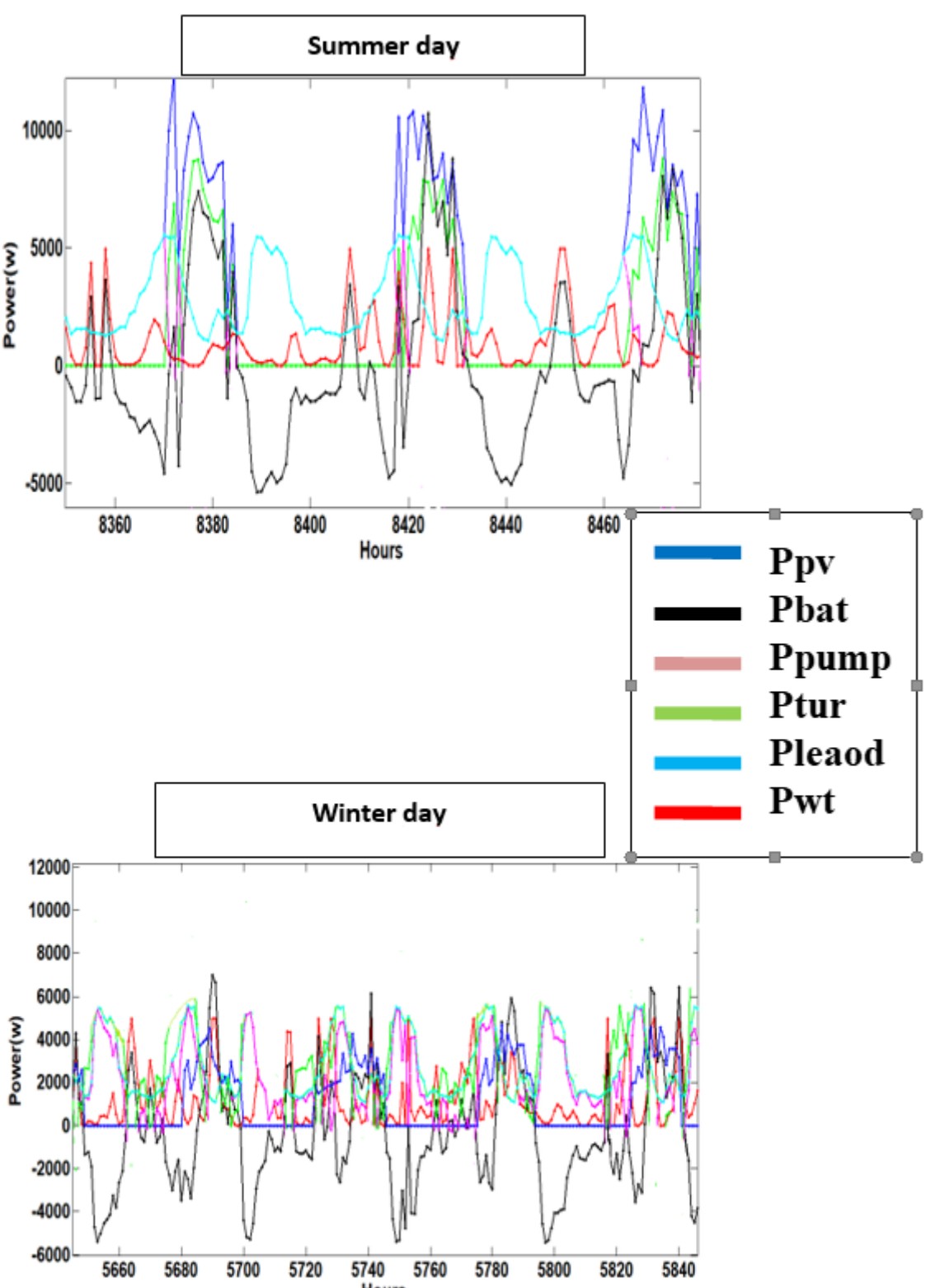

**Figure 19.** Power evolution for two typical days: one in summer and the other in winter.

In the nocturnal days and when the wind is strong, it is very clear that the supply of the load is assured by the wind turbine and the discharge of the storage system.

Finally, the state of the charge of the tank is illustrated in Figure 20. We notice that the reservoir is charged and discharged with a SOC varying between 100% and 50%.Thisrespects the optimization constraints that we have set.

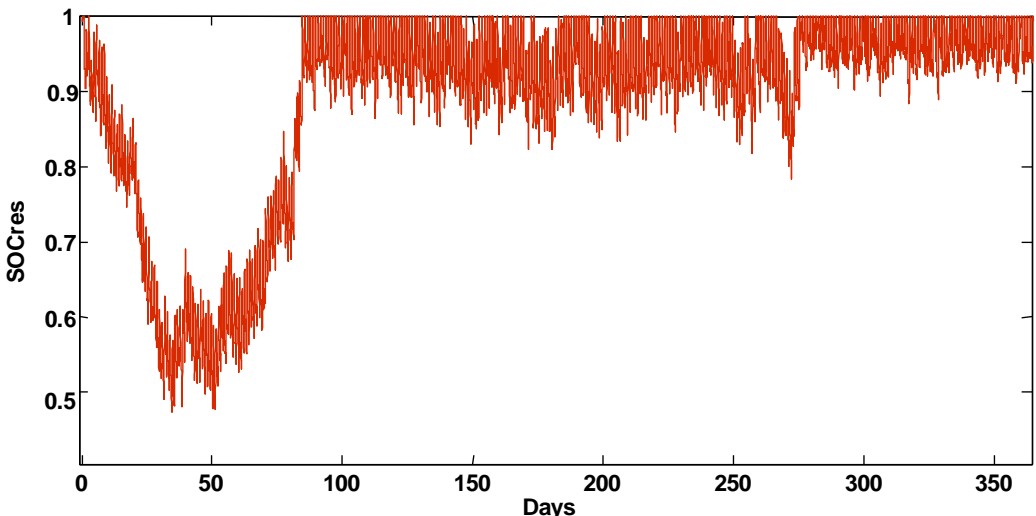

**Figure 20.** State of charge of the tank.

## 6. Conclusions

This paper studies a hybrid wind–photovoltaic system associated with an electrochemical and hydraulic storage system. The objective of this system is to produce electricity to satisfy the demand of a consumer on a remote island in Tunisia at any time. From there, we presented the components of the studied system and their principles of operation. Then we gave their mathematical models. For the photovoltaic production chain, we presented two types of modeling of the PV panels: a fine modeling that faithfully reproduces the behaviors of these panels and a simplified energy modeling that will be integrated in the optimization process. Concerning the wind generation chain, we created a wind profile using the probability function of the wind speed (the Weibull distribution) for every half hour for one year. Then, we modeled the wind power under different wind speeds. For the electrochemical and hydraulic storage system, based on the electrical model and its equations, we determined the different parameters that characterize the battery and the reservoir and that allow us to determine the power and the states of charge of the battery (SOCbat) and reservoir (SOCres).

The bi-objective optimization methodology (LPSP-Cost), using an evolutionary algorithm NSGAII, was used to size the optimal system size, i.e., to find the (NPV, NWT, Nbat, V, and Hm), which satisfies the consumer demand and minimizes the cost of the system, while ensuring an honorable lifetime for the batteries. Results from multi-objective optimization using the NSGAII genetic algorithm were presented and discussed. We select three optimal configurations to analyze. The results show the effectiveness and robustness of our synthesis approach to ensure the continued electrification of the remote island.

To finish, several perspectives can be considered in order to complete the experimental validation, such as the use of weather forecasts based on intelligent algorithms. In addition to wind and photovoltaic, the use of an energy source (such as the fuel cell) is very interesting to make the system more efficient during the worst periods of the year.

**Author Contributions:** Conceptualization, D.M. and C.C.; Formal analysis, M.H.; Investigation, M.H.; Methodology, A.M.; Supervision, C.C. and D.M.; Writing—original draft, M.H. All authors have read and agreed to the published version of the manuscript.

**Funding:** This research received no external funding.

**Acknowledgments:** This work was supported by the Tunisian Ministry of High Education, Research and Technology. I would like to express my deep gratitude to Malek Belouda for his help in doing the data analysis.

**Conflicts of Interest:** The authors declare no conflict of interest.

## Nomenclature

| | |
|---|---|
| **Pload** | Load power (w) |
| **PPV** | Photovoltaic power (w) |
| **PWT** | Electrical wind turbine output powe(W) |
| **Hm** | Head height (m) |
| **P** | Water density (1000 kg/m$^3$) |
| **Q** | Electron charge (C) |
| **Rcel** | Internal resistance of a cell($\Omega$) |
| **Ppump** | Pumping power (W) |
| **Ptur** | Turbine power (W) |
| **Q** | Electron charge (C) |
| **Q** | Water volumetric flow (m$^3$/s) |
| **Ccel** | Cell capacity (A.h) |
| **Cp** | Power coefficient |
| **Eg** | Band gap energy (ev) |
| **G** | Real solar irradiation (W/m$^2$) |
| **Ibat** | Battery current (A) |
| **Icel** | Cell current (A) |
| **Id** | Diode current (A) |
| **Iph** | Photo-current (A) |
| **Is** | Diode saturation current (A) |
| **K** | Boltzmann's constant (J/K) |
| **N** | Diode ideality factor |
| **Np** | Number of PV modules in parallel |
| **Ns** | Number of PV cells in series |
| **Pbat** | Battery power (w) |
| **Rs** | Series resistance ($\Omega$) |
| **Rsh** | Parallel resistance ($\Omega$) |
| **SOC** | State of charge of the reservoir |
| **SPV** | PV panel surface (m$^2$) |
| **Tc** | Cell temperature (K) |
| **Ts** | Sampling time (hours) |
| **V** | Wind speed (m/s) |
| **Vcel** | Voltage at cell terminals (V) |
| **VT** | Thermodynamic potential (V) |
| **U** | Voltage of cell (V) |
| **ηinv** | Inverter efficiency |
| **ηPV** | PV system efficiency |

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
