# Peer review of "Technical, Economic, and Intelligent Optimization for the Optimal Sizing of a Hybrid Renewable Energy System with a Multi Storage System on Remote Island in Tunisia"

_electronics, doi:10.3390/electronics11203261_

Round 1

Reviewer 1 Report

This manuscript presents a multiobjective optimization using an evolutionary algorithm to extract the optimal sizing choice of the hybrid renewable energy system of a remote island. I tried to read carefully from the Abstract to the Conclusion part, and I guess the novelty of the manuscript is the bi-objective optimization methodology named “loss of power supply probability (LPSP)” using a non-dominated sorting genetic algorithm (NSGA)-II. Finally, the authors conclude that the results show the effectiveness and robustness of your synthesis approach to ensure the continued electrification of the remote island.

However, which scenarios are used to justify the above conclusion?

Moreover, the effectiveness of the proposed method should be expressed by a detailed number. For example, in the paper with the same authors of [25] in the Applied Energy journal, the authors pointed out clearly the Levelized cost of energy (COE) of their system is $0.595/kW. So, in your study, which is the effective number you obtained in this study?

1. Please fix all the error grammar. I saw so many typos in this manuscript.

2. The order of references is not reasonable.

3. The title of the paper has several spelling mistakes. For example, the “determinate” word and etc. Generally, presentation and English need to be revised very very carefully. Even if in some paragraphs, the authors did not translate completely from French to English.

Author Response

We would like to thank you for taking the necessary time and effort to review the manuscript. We sincerely appreciate all your valuable comments and suggestions, which helped us in improving the quality of the manuscript. Please see below, in blue, for a point-by-point response to your comments and the revised version of the manuscript.

 Comment 1: which scenarios are used to justify the conclusion?

Response: The objective of the approach adopted is not to constitute an automated design tool, but rather to provide design assistance through a set of results facilitating the sizing choices of the system.  Our work is concerned with optimizing the elements of this system size by relying on different criteria, the crucial phase will consist in the definition of these problem optimization criteria. We chose the NSGA-II as the optimization method for solving the design problem. This algorithm has become an essential reference in the field of multi-criteria optimization by genetic algorithms.

We analyze the characteristics of three different configurations. The analysis proved a good indicator for choosing the right configuration for a given application.

 Comment 2:  The effectiveness of the proposed method should be expressed by a detailed number. For example, in the paper with the same authors of [25] in the Applied Energy journal, the authors pointed out clearly the Levelized cost of energy (COE) of their system is $0.595/kW. So, in your study, which is the effective number you obtained in this study?

Response: In our study, we estimate the cost of energy (cost per Khw) produced by our system through the costs of PV panels, wind turbines, converters, installation, maintenance, and replacement of components over a cycle of life (Tcycle) equal to 25 years. Cost modeling is done through energy models that relate the cost of each element to its average power. For the PV system, the cost is esteemed at 2000 €/kW. This includes all the equipment and accessories needed to operate the PV panels (PV panels, MPPT static converters, converters, control circuits, sensors, and so forth. For the pumping system, the cost is esteemed at 240 €/kW. For the turbine system, the cost is esteemed at 50 €/kW. 

 Comment 3:  The order of references is not reasonable.

Response: The references have been updated.

 Comment 4:  The title of the paper has several spelling mistakes.

Response: Title has been updated, such that” Technical, Economic and Intelligent Optimization for Optimal Sizing of  Hybrid Renewable Energy System with Multi Storage System in Remote Island in Tunisia”.

Reviewer 2 Report

This topic is widely studied and new paper have to stress what their specific contribution is to the research field.
First of all a clear analysis of the state of the art should be reported, avoiding the use of lumped references and with the specific aim of putting in evidence what is the expected contribution of present work.
There are numerous self-citations whereas there is a lack of reference to the wide amount of recent scientific literature that can be found on this specific topic.
English language should also be improved as in many different parts the paper is difficult to understand.
The choice of the different models should be supported by specific considerations. The model used are rather simply and their choice needs explanation.
Many results presented are very sensitive the the specific geographical conditions analyzed. It would be important to address the general application of the present work to other environmental conditions.

Author Response

We would like to thank you for taking the necessary time and effort to review the manuscript. We sincerely appreciate all your valuable comments and suggestions, which helped us in improving the quality of the manuscript. Please see below, in blue, for a point-by-point response to your comments and the revised version of the manuscript.

 Comment 1:  This topic is widely studied and new papers have to stress what their specific contribution is to the research field.

Response:  Optimal sizing of PV /Wind/ hydraulic/Battery with multi-objective formulation with a combination of cost-LPSP is rarely found in the literature. This is the first attempt to apply the proposed optimization techniques, to optimize the system components, for solving a real problem of power shortage on a remote island located in Tunisia, based on real-time meteorological data of the site for developing this promising remote area. Moreover, the energetic models proposed in this paper, for determining the best optimal size of the components of the hybrid energy system, are novel and different from the conventional models, leading to notable improvements in the optimization problems.

 Comment 2: First of all a clear analysis of the state of the art should be reported, avoiding the use of lumped references and with the specific aim of putting in evidence what is the expected contribution of the present work:

Response:

The state of the art is reported on pages 1 and 2.

 Comment 3:  There are numerous self-citations whereas there is a lack of reference to the wide amount of recent scientific literature that can be found on this specific topic.

Response: We have added recent scientific literature to the manuscript that can be found in “Introduction and background of this research”.

 Comment 4:  The choice of the different models should be supported by specific considerations. The models used are rather simple and their choice needs explanation

Response: In the context of an integrated optimal design, the processing time cost is of high priority. Therefore, the use of energetic models is unavoidable in such optimization problems.

 Comment 5:  Many results presented are very sensitive to the specific geographical conditions analyzed. It would be important to address the general application of the present work to other environmental conditions.

Response: We agree that the results presented are very sensitive to the specific geographical. Our hybrid system has been chosen because of its reliability and its simple architecture. The proposed study takes account of stochastic features of solar irradiation and wind speed in a particular location in Tunisia with given deterministic power demand. Based on the developed energetic models the considered PV/WT/hydraulic/batteries system was solved with an evolutionary algorithm using an NSGAII multi-objective algorithm.

Reviewer 3 Report

The subject of the article is remarkable. Today, it is important to deal with the issue of meeting the energy needs of isolated islands efficiently from renewable resources. Minor revision is required to make the following corrections to the article. I believe that minor revision will increase the scientific quality of the article and make it ready for publication.

There are errors in spelling and punctuation, they should be reviewed.

Paragraph 2 on page 4 should start with a capital letter. while writing the electron charge on the same page, it should be 1.6.10e-19 C.

When talking about PV modules, it should be explained that losses due to environmental factors (shading, thermal efficiency, etc.) that are frequently encountered reduce their efficiency. The effectiveness of environmental factors should be explained by referring to these sources: DOI: 10.2339/politeknik.903989, https://www.researchgate.net/publication/341204359_PHOTOVOLTAIC_SYSTEMS_A_REVIEW, https://www.researchgate.net/publication/357017737_THERMAL_ANALYSIS_OF_SEMI-TRANSPARENT_PHOTOVOLTAIC_WINDOW_DESIGN_AND_CONVENTIONAL_DOUBLE_GLAZING_WINDOW_MODEL, https://www.researchgate.net/publication/351811591_PERFORMANCE_PREDICTION_APPROACH_USING_RAINFALL_BASED_ON_ARTIFICIAL_NEURAL_NETWORK_FOR_PV_MODULE, https://www.researchgate.net/publication/345156756_FREQUENT_FAULTS_ON_THE_DC_SIDE_IN_PHOTOVOLTAIC.

3rd paragraph on page 7 is not written in English. It needs to be corrected. There are also non-English texts on the same page.

The font size of the 11th and 13th equations is small. It should be made in accordance with the writing template of the journal.

On page 9, "2" should be written as a superscript in the square meter expression.

The text in the g) heading on page 10 is missing.

It has been stated that the thermal and aging effects of the batteries are neglected. This paragraph should be expanded to indicate the possible consequences of thermal and aging effects. Thus, it will be understood how important the neglected situation really is.

Table 2 contains French words. English equivalents should be written.

On page 13: The sentence "For the PV system (the cost combine the PV panels, converters, cables, sensors and control circuits is estimed at 2000 €/kW(CostPV).") is not correct in structure. Parentheses should be corrected and turned into a complete sentence .. In addition, the above-mentioned sources should be included in the article and maintenance and operating costs should be added to the sentence.

There is an itemization error in reference 37.

Author Response

We would like to take this opportunity to thank you for the effort and expertise that you contributed to reviewing the article, without which it would be impossible to maintain the high standards of peer-reviewed journals. Please see below, in blue, for a point-by-point response to your comments and the revised version of the manuscript.

Comment 1 When talking about PV modules, it should be explained that losses due to environmental factors (shading, thermal efficiency, etc.) that are frequently encountered reduce their efficiency.

Response: Thank you for this suggestion. We have added the suggested content to the manuscript on page 5,” PV module modeling”

Comment 2: The text in the g) heading on page 10 is missing.

Response: The text in the g) has been updated.

Comment 3: It has been stated that the thermal and aging effects of the batteries are neglected. This paragraph should be expanded to indicate the possible consequences of thermal and aging effects. Thus, it will be understood how important the neglected situation really is.

Response: As suggested by the reviewer, we have added the suggested content to the manuscript in the heading on page 10.

Comment 4: On page 13: The sentence "For the PV system (the cost combine the PV panels, converters, cables, sensors and control circuits is estimed at 2000 €/kW(CostPV).") is not correct in structure. Parentheses should be corrected and turned into a complete sentence.. In addition, the above-mentioned sources should be included in the article and maintenance and operating costs should be added to the sentence.

Response: Thank you for pointing this out. The sentence is corrected “The cost, combining the PV panels, converters, cables, sensors and control circuits, is esteemed at 2000 €/kW (CostPV)”.

Comment 4: There is an itemization error in reference 37.

Response: Reference 37 has been updated.

Additional clarifications

In addition to the above comments, all spelling and grammatical errors pointed out by the reviewers have been corrected.

We look forward to hearing from you in due time regarding our submission and to responding to any further questions and comments you may have.

Sincerely,

Round 2

Reviewer 2 Report

Comment 4:  The choice of the different models should be supported by specific considerations. The models used are rather simple and their choice needs explanation

Response: In the context of an integrated optimal design, the processing time cost is of high priority. Therefore, the use of energetic models is unavoidable in such optimization problems.

Reviewer comment: This statement should be supported by data, e.g. what is the calculation time advantage of this type of simple models vs. the reduced predictive capability of simple models.

I Aknowledge authors efforts in answering the review points.